# Flexible Neural Representation for Physics Prediction

**Damian Mrowca**[1,*]**, Chengxu Zhuang**[2,*]**, Elias Wang**[3,*]**, Nick Haber**[2,4,5]** , Li Fei-Fei**[1] **,**
**Joshua B. Tenenbaum**[7,8] **, and Daniel L. K. Yamins**[1,2,6]

Department of Computer Science[1], Psychology[2], Electrical Engineering[3], Pediatrics[4] and
Biomedical Data Science[5], and Wu Tsai Neurosciences Institute[6], Stanford, CA 94305
Department of Brain and Cognitive Sciences[7], and Computer Science and Artificial Intelligence
Laboratory[8], MIT, Cambridge, MA 02139

{mrowca, chengxuz, eliwang}@stanford.edu

## Abstract

Humans have a remarkable capacity to understand the physical dynamics of objects
in their environment, flexibly capturing complex structures and interactions at
multiple levels of detail. Inspired by this ability, we propose a hierarchical particle-
based object representation that covers a wide variety of types of three-dimensional
objects, including both arbitrary rigid geometrical shapes and deformable materi-
als. We then describe the Hierarchical Relation Network (HRN), an end-to-end
differentiable neural network based on hierarchical graph convolution, that learns
to predict physical dynamics in this representation. Compared to other neural
network baselines, the HRN accurately handles complex collisions and nonrigid
deformations, generating plausible dynamics predictions at long time scales in
novel settings, and scaling to large scene configurations. These results demonstrate
an architecture with the potential to form the basis of next-generation physics
predictors for use in computer vision, robotics, and quantitative cognitive science.

## 1   Introduction

Humans efficiently decompose their environment into objects, and reason effectively about the
dynamic interactions between these objects [43, 45]. Although human intuitive physics may be
quantitatively inaccurate under some circumstances [32], humans make qualitatively plausible guesses
about dynamic trajectories of their environments over long time horizons [41]. Moreover, they either
are born knowing, or quickly learn about, concepts such as object permanence, occlusion, and
deformability, which guide their perception and reasoning [42].

An artificial system that could mimic such abilities would be of great use for applications in computer
vision, robotics, reinforcement learning, and many other areas. While traditional physics engines
constructed for computer graphics have made great strides, such routines are often hard-wired
and thus challenging to integrate as components of larger learnable systems. Creating end-to-end
differentiable neural networks for physics prediction is thus an appealing idea. Recently, Chang et al.
[11] and Battaglia et al. [4] have illustrated the use of neural networks to predict physical object
interactions in (mostly) 2D scenarios by proposing object-centric and relation-centric representations.
Common to these works is the treatment of scenes as graphs, with nodes representing object point
masses and edges describing the pairwise relations between objects (e.g. gravitational, spring-like, or
repulsing relationships). Object relations and physical states are used to compute the pairwise effects
between objects. After combining effects on an object, the future physical state of the environment is
predicted on a per-object basis. This approach is very promising in its ability to explicitly handle

---

object interactions. However, a number of challenges have remained in generalizing this approach to real-world physical dynamics, including representing arbitrary geometric shapes with sufficient resolution to capture complex collisions, working with objects at different scales simultaneously, and handling non-rigid objects of nontrivial complexity.

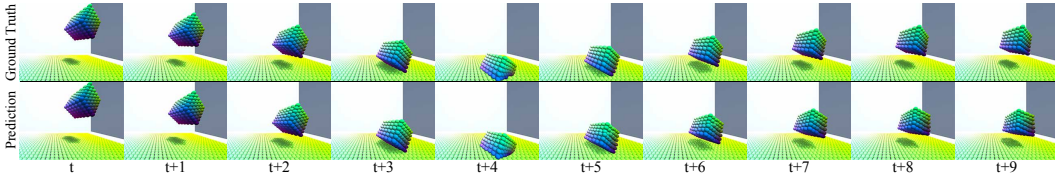

Figure 1: **Predicting physical dynamics.** Given past observations the task is to predict the future physical state of a system. In this example, a cube deforms as it collides with the ground. The top row shows the ground truth and the bottom row the prediction of our physics prediction network.

Several of these challenges are illustrated in the fast-moving deformable cube sequence depicted in Figure 1. Humans can flexibly vary the level of detail at which they perceive such objects in motion: The cube may naturally be conceived as an undifferentiated point mass as it moves along its initial kinematic trajectory. But as it collides with and bounces up from the floor, the cube's complex rectilinear substructure and nonrigid material properties become important for understanding what happens and predicting future interactions. The ease with which the human mind handles such complex scenarios is an important explicandum of cognitive science, and also a key challenge for artificial intelligence. Motivated by both of these goals, our aim here is to develop a new class of neural network architectures with this human-like ability to reason flexibly about the physical world.

To this end, it would be natural to extend the interaction network framework by representing each object as a (potentially large) set of connected particles. In such a representation, individual constituent particles could move independently, allowing the object to deform while being constrained by pairwise relations preventing the object from falling apart. However, this type of particle-based representation introduces a number of challenges of its own. Conceptually, it is not immediately clear how to efficiently propagate effects across such an object. Moreover, representing every object with hundreds or thousands of particles would result in an exploding number of pairwise relations, which is both computationally infeasible and cognitively unnatural.

As a solution to these issues, we propose a novel cognitively-inspired hierarchical graph-based object representation that captures a wide variety of complex rigid and deformable bodies (Section 3), and an efficient hierarchical graph-convolutional neural network that learns physics prediction within this representation (Section 4). Evaluating on complex 3D scenarios, we show substantial improvements relative to strong baselines both in quantitative prediction accuracy and qualitative measures of prediction plausibility, and evidence for generalization to complex unseen scenarios (Section 5).

## 2 Related Work

An efficient and flexible predictor of physical dynamics has been an outstanding question in neural network design. In computer vision, modeling moving objects in images or videos for action recognition, future prediction, and object tracking is of great interest. Similarly in robotics, action-conditioned future prediction from images is crucial for navigation or object interactions. However, future predictors operating directly on 2D image representations often fail to generate sharp object boundaries and struggle with occlusions and remembering objects when they are no longer visually observable [1, 17, 16, 28, 29, 33, 34, 19]. Representations using 3D convolution or point clouds are better at maintaining object shape [46, 47, 10, 36, 37], but do not entirely capture object permanence, and can be computationally inefficient. More similar to our approach are inverse graphics methods that extract a lower dimensional physical representation from images that is used to predict physics [25, 26, 51, 50, 52, 53, 7, 49]. Our work draws inspiration from and extends that of Chang et al. [11] and Battaglia et al. [4], which in turn use ideas from graph-based neural networks [39, 44, 9, 30, 22, 14, 13, 24, 8, 40]. Most of the existing work, however, does not naturally handle complex scene scenarios with objects of widely varying scales or deformable objects with complex materials.

Physics simulation has also long been studied in computer graphics, most commonly for rigid-body collisions [2, 12]. Particles or point masses have also been used to represent more complex physical

objects, with the neural network-based NeuroAnimator being one of the earliest examples to use a hierarchical particle representation for objects to advance the movement of physical objects [18]. Our particle-based object representation also draws inspiration from recent work on (non-neural-network) physics simulation, in particular the NVIDIA FleX engine [31, 6]. However, unlike this work, our solution is an end-to-end differentiable neural network that can learn from data.

Recent research in computational cognitive science has posited that humans run physics simulations in their mind [5, 3, 20, 48, 21]. It seems plausible that such simulations happen at just the right level of detail which can be flexibly adapted as needed, similar to our proposed representation. Both the ability to imagine object motion as well as to flexibly decompose an environment into objects and parts form an important prior that humans rely on for further learning about new tasks, when generalizing their skills to new environments or flexibly adapting to changes in inputs and goals [27].

## 3 Hierarchical Particle Graph Representation

A key factor for predicting the future physical state of a system is the underlying representation used. A simplifying, but restrictive, often made assumption is that all objects are rigid. A rigid body can be represented with a single point mass and unambiguously situated in space by specifying its position and orientation, together with a separate data structure describing the object's shape and extent. Examples are 3D polygon meshes or various forms of 2D or 3D masks extracted from perceptual data [10, 16]. The rigid body assumption describes only a fraction of the real world, excluding, for example, soft bodies, cloths, fluids, and gases, and precludes objects breaking and combining. However, objects are divisible and made up of a potentially large numbers of smaller sub-parts.

Given a scene with a set of objects $O$, the core idea is to represent each object $o \in O$ with a set of particles $P_o \equiv \{p_i | i \in o\}$. Each particle's state at time $t$ is described by a vector in $\mathbb{R}^7$ consisting of its position $x \in \mathbb{R}^3$, velocity $\delta \in \mathbb{R}^3$, and mass $m \in \mathbb{R}^+$. We refer to $p_i$ and this vector interchangeably.

Particles are spaced out across an object to fully describe its volume. In theory, particles can be arbitrarily placed within an object. Thus, less complex parts can be described with fewer particles (e.g. 8 particles fully define a cube). More complicated parts (e.g. a long rod) can be represented with more particles. We define $P$ as the set $\{p_i | 1 \le i \le N_P\}$ of all $N_P$ particles in the observed scene.

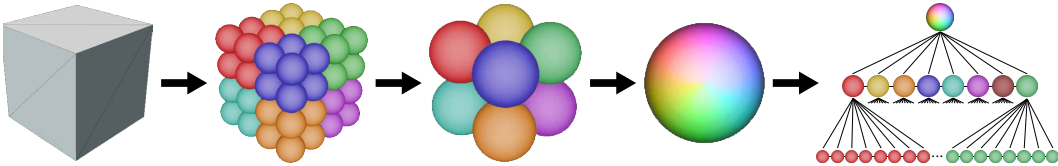

Figure 2: **Hierarchical graph-based object representation**. An object is decomposed into particles. Particles (of the same color) are grouped into a hierarchy representing multiple object scales. Pairwise relations constrain particles in the same group and to ancestors and descendants.

To fully physically describe a scene containing multiple objects with particles, we also need to define how the particles relate to each other. Similar to Battaglia et al. [4], we represent relations between particles $p_i$ and $p_j$ with $K$-dimensional pairwise relationships $R = \{r_{ij} \in \mathbb{R}^K\}$. Each relationship $r_{ij}$ within an object encodes material properties. For example, for a soft body $r_{ij} \in \mathbb{R}$ represents the local material stiffness, which need not be uniform within an object. Arbitrarily-shaped objects with potentially nonuniform materials can be represented in this way. Note that the physical interpretation of $r_{ij}$ is learned from data rather than hard-coded through equations. Overall, we represent the scene by a node-labeled graph $G = \langle P, R \rangle$ where the particles form the nodes $P$ and the relations define the (directed) edges $R$. Except for the case of collisions, different objects are disconnected components within $G$.

The graph $G$ is used to propagate effects through the scene. It is infeasible to use a fully connected graph for propagation as pairwise-relationship computations grow with $\mathcal{O}(N_P^2)$. To achieve $\mathcal{O}(N_P log(N_P))$ complexity, we construct a *hierarchical scene (di)graph $G_H$* from $G$ in which the nodes of each connected component are organized into a tree structure: First, we initialize the leaf nodes $L$ of $G_H$ as the original particle set $P$. Then, we extend $G_H$ by a root node for each connected component (object) in $G$. The root node states are defined as the aggregates of their leaf node states. The root nodes are connected to their leaves with directed edges and vice versa.

At this point, $G_H$ consists of the leaf particles $L$ representing the finest scene resolution and one root node for each connected component describing the scene at the object level. To obtain intermediate levels of detail, we then cluster the leaves $L$ in each connected component into smaller subcomponents using a modified k-means algorithm. We add one node for each new subcomponent and connect its leaves to the newly added node and vice versa. This newly added node is then labeled as the direct ancestors for its leaves and its leaves are siblings to each other. We then connect the added intermediate nodes with each other if and only if their respective subcomponent leaves are connected. Lastly, we add directed edges from the root node of each connected component to the new intermediate nodes in that component, and remove edges between leaves not in the same cluster. The process then recurses within each new subcomponent. See Algorithm 1 in the supplementary for details.

We denote the sibling(s) of a particle $p$ by $\text{sib}(p)$, its ancestor(s) by $\text{anc}(p)$, its parent by $\text{par}(p)$, and its descendant(s) by $\text{des}(p)$. We define $\text{leaves}(p_a) = \{p_l \in L \mid p_a \in \text{anc}(p_l)\}$. Note that in $G_H$, directed edges connect $p_i$ and $\text{sib}(p_i)$, leaves $p_l$ and $\text{anc}(p_l)$, and $p_i$ and $\text{des}(p_i)$; see Figure 3b.

# 4 Physics Prediction Model

In this section we introduce our physics prediction model. It is based on hierarchical graph convolution, an operation which propagates relevant physical effects through the graph hierarchy.

## 4.1 Hierarchical Graph Convolutions For Effect Propagation

In order to predict the future physical state, we need to resolve the constraints that particles connected in the hierarchical graph impose on each other. We use graph convolutions to compute and propagate these effects. Following Battaglia et al. [4], we implement a *pairwise graph convolution* using two basic building blocks: (1) A pairwise processing unit $\phi$ that takes the sender particle state $p_s$, the receiver particle state $p_r$ and their relation $r_{sr}$ as input and outputs the effect $e_{sr} \in \mathbb{R}^E$ of $p_s$ on $p_r$, and (2) a commutative aggregation operation $\Sigma$ which collects and computes the overall effect $e_r \in \mathbb{R}^E$. In our case, this is a simple summation over all effects on $p_r$. Together these two building blocks form a convolution on graphs as shown in Figure 3a.

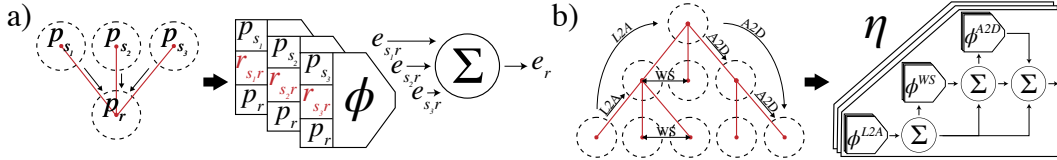

Figure 3: **Effect propagation through graph convolutions. a) Pairwise graph convolution** $\phi$. A receiver particle $p_r$ is constrained in its movement through graph relations $r_{sr}$ with sender particle(s) $p_s$. Given $p_s$, $p_r$ and $r_{sr}$, the effect $e_{sr}$ of $p_s$ on $p_r$ is computed using a fully connected neural network. The overall effect $e_r$ is the sum of all effects on $p_r$. **b) Hierarchical graph convolution** $\eta$. Effects in the hierarchy are propagated in three consecutive steps. (1) $\phi_{L2A}$. Leaf particles $L$ propagate effects to all of their ancestors $A$. (2) $\phi_{WS}$. Effects are exchanged between siblings $S$. (3) $\phi_{A2D}$. Effects are propagated from the ancestors $A$ to all of their descendants $D$.

Pairwise processing limits graph convolutions to only propagate effects between directly connected nodes. For a generic flat graph, we would have to repeatedly apply this operation until the information from all particles has propagated across the whole graph. This is infeasible in a scenario with many particles. Instead, we leverage direct connections between particles and their ancestors in our hierarchy to propagate all effects across the entire graph in *one* model step. We introduce a *hierarchical graph convolution*, a three stage mechanism for effect propagation as seen in Figure 3b:

The first *L2A (Leaves to Ancestors) stage* $\phi^{L2A}(p_l, p_a, r_{la}, e_l^0)$ predicts the effect $e_{la}^{L2A} \in \mathbb{R}^E$ of a leaf particle $p_l$ on an ancestor particle $p_a \in \text{anc}(p_l)$ given $p_l$, $p_a$, the material property information of $r_{la}$, and input effect $e_l^0$ on $p_l$. The second *WS (Within Siblings) stage* $\phi^{WS}(p_i, p_j, r_{ij}, e_i^{L2A})$ predicts the effect $e_{ij}^{WS} \in \mathbb{R}^E$ of sibling particle $p_i$ on $p_j \in \text{sib}(p_i)$. The third *A2D (Ancestors to Descendants) stage* $\phi^{A2D}(p_a, p_d, r_{ad}, e_a^{L2A} + e_a^{WS})$ predicts the effect $e_{ij}^{A2D} \in \mathbb{R}^E$ of an ancestor particle $p_a$ on a descendant particle $p_d \in \text{des}(p_a)$. The total propagated effect $e_i$ on particle $p_i$ is

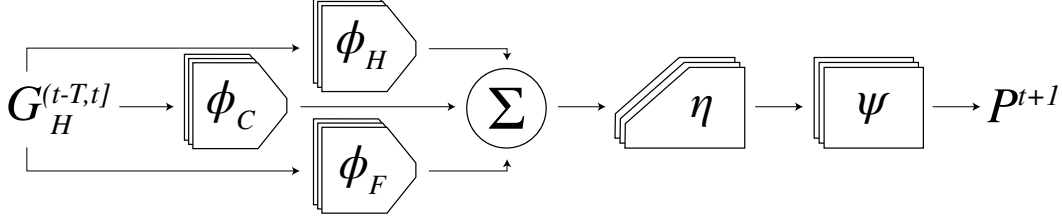

Figure 4: **Hierarchical Relation Network**. The model takes the past particle graphs $G_H^{(t-T,t]} = \langle P^{(t-T,t]}, R^{(t-T,t]} \rangle$ as input and outputs the next states $P^{t+1}$. The inputs to each graph convolutional effect module $\phi$ are the particle states and relations, the outputs the respective effects. $\phi_H$ processes past states, $\phi_C$ collisions, and $\phi_F$ external forces. The hierarchical graph convolutional module $\eta$ takes the sum of all effects, the pairwise particle states, and relations and propagates the effects through the graph. Finally, $\psi$ uses the propagated effects to compute the next particle states $P^{t+1}$.

computed by summing the various effects on that particle, $e_i = e_i^{L2A} + e_i^{WS} + e_i^{A2D}$ where

$$e_a^{L2A} = \sum_{p_l \in \text{leaves}(p_a)} \phi^{L2A}(p_l, p_a, r_{la}, e_l^0) \qquad e_j^{WS} = \sum_{p_i \in \text{sib}(p_j)} \phi^{WS}(p_i, p_j, r_{ij}, e_i^{L2A})$$

$$e_d^{A2D} = \sum_{p_a \in \text{anc}(p_d)} \phi^{A2D}(p_a, p_d, r_{ad}, e_a^{L2A} + e_a^{WS}).$$

In practice, $\phi^{L2A}, \phi^{WS}$, and $\phi^{A2D}$ are realized as fully-connected networks with shared weights that receive an additional ternary input (0 for L2A, 1 for WS, and 2 for A2D) in form of a one-hot vector.

Since all particles within one object are connected to the root node, information can flow across the entire hierarchical graph in at most two propagation steps. We make use of this property in our model.

## 4.2 The Hierarchical Relation Network Architecture

This section introduces the *Hierarchical Relation Network* (HRN), a neural network for predicting future physical states shown in Figure 4. At each time step $t$, HRN takes a history of $T$ previous particle states $P^{(t-T,t]}$ and relations $R^{(t-T,t]}$ in the form of hierarchical scene graphs $G_H^{(t-T,t]}$ as input. $G_H^{(t-T,t]}$ dynamically changes over time as directed, unlabeled virtual collision relations are added for sufficiently close pairs of particles. HRN also takes external effects on the system (for example gravity $g$ or external forces $F$) as input. The model consists of three pairwise graph convolution modules, one for external forces ($\phi_F$), one for collisions ($\phi_C$) and one for past states ($\phi_H$), followed by a hierarchical graph convolution module $\eta$ that propagates effects through the particle hierarchy. A fully-connected module $\psi$ then outputs the next states $P^{t+1}$.

In the following, we briefly describe each module. For ease of reading we *drop the notation $(t-T,t]$* and assume that all variables are subject to this time range unless otherwise noted.

**External Force Module** The *external force module* $\phi_F$ converts forces $F \equiv \{f_i\}$ on leaf particles $p_i \in P^L$ into effects $\phi_F(p_i, f_i) = e_i^F \in \mathbb{R}^E$.

**Collision Module** Collisions between objects are handled by dynamically defining pairwise collision relations $r_{ij}^C$ between leaf particles $p_i \in P^L$ from one object and $p_j \in P^L$ from another object that are close to each other [11]. The *collision module* $\phi_C$ uses $p_i$, $p_j$ and $r_{ij}^C$ to compute the effects $\phi_C(p_j, p_i, r_{ij}^C) = e_{ji}^C \in \mathbb{R}^E$ of $p_j$ on $p_i$ and vice versa. With $d^t(i,j) = \|x_i^t - x_j^t\|$, the overall collision effects equal $e_i^C = \sum_j \{e_{ji} | d^t(i,j) < D_C\}$. The hyperparameter $D_C$ represents the maximum distance for a collision relation.

**History Module** The *history module* $\phi_H$ predicts the effects $\phi(p_i^{(t-T,t-1]}, p_i^t) \in e_i^H$ from past $p_i^{(t-T,t-1]} \in P^L$ on current leaf particle states $p_i^t \in P^L$.

**Hierarchical Effect Propagation Module** The *hierarchical effect propagation module* $\eta$ propagates the overall effect $e_i^0 = e_i^F + e_i^C + e_i^H$ from external forces, collisions and history on $p_i$ through the particle hierarchy. $\eta$ corresponds to the three-stage hierarchical graph convolution introduced in

Figure 3 b) which given the pairwise particle states $p_i$ and $p_j$, their relation $r_{ij}$, and input effects $e_i^0$, outputs the total propagated effect $e_i$ on each particle $p_i$.

**State Prediction Module**   We use a simple fully-connected network $\psi$ to predict the next particle states $P^{t+1}$. In order to get more accurate predictions, we leverage the hierarchical particle representation by predicting the dynamics of any given particle within the local coordinate system originated at its parent. The only exceptions are object root particles for which we predict the global dynamics. Specifically, the *state prediction module* $\psi(g, p_i, e_i)$ predicts the local future delta position $\delta_{i,\ell}^{t+1} = \delta_i^{t+1} - \delta_{\text{par}(i)}^{t+1}$ using the particle state $p_i$, the total effect $e_i$ on $p_i$, and the gravity $g$ as input. As we only predict global dynamics for object root particles, the gravity is only applied to these root particles. The final future delta position in world coordinates is computed from local information as $\delta_i^{t+1} = \delta_{i,\ell}^{t+1} + \sum_j \delta_{j,\ell}^{t+1}, j \in \text{anc}(i)$.

### 4.3   Learning Physical Constraints through Loss Functions and Data

Traditionally, physical systems are modeled with equations providing fixed approximations of the real world. Instead, we choose to learn physical constraints, including the meaning of the material property vector, from data. The error signal we found to work best is a combination of three objectives. (1) We predict the position change $\delta_{i,\ell}^{t+1}$ between time step $t$ and $t+1$ independently for all particles in the hierarchy. In practice, we find that $\delta_{i,\ell}^{t+1}$ will differ in magnitude for particles in different levels. Therefore, we normalize the local dynamics using the statistics from all particles in the same level (*local loss*). (2) We also require that the global future delta position $\delta_i^{t+1}$ is accurate (*global loss*). (3) We aim to preserve the intra-object particle structure by imposing that the pairwise distance between two connected particles $p_i$ and $p_j$ in the next time step $d^{t+1}(i,j)$ matches the ground truth. In the case of a rigid body this term works to preserve the distance between particles. For soft bodies, this objective ensures that pairwise local deformations are learned correctly (*preservation loss*).

The total objective function linearly combines (1), (2), and (3) weighted by hyperparameters $\alpha$ and $\beta$:

$$Loss = \alpha \Big( \sum_{p_i} \|\hat{\delta}_{i,\ell}^{t+1} - \delta_{i,\ell}^{t+1}\|^2 + \beta \sum_{p_i} \|\hat{\delta}_i^{t+1} - \delta_i^{t+1}\|^2 \Big) + (1-\alpha) \sum_{p_i \in \text{sib}(p_j)} \|\hat{d}^{t+1}(i,j) - d^{t+1}(i,j)\|^2$$

## 5   Experiments

In this section, we examine the HRN's ability to accurately predict the physical state across time in scenarios with rigid bodies, deformable bodies (soft bodies, cloths, and fluids), collisions, and external actions. We also evaluate the generalization performance across various object and environment properties. Finally, we present some more complex scenarios including (e.g.) falling block towers and dominoes. Prediction roll-outs are generated by recursively feeding back the HRN's one-step prediction as input. We strongly encourage readers to have a look at result examples shown in main text figures, supplementary materials, and at https://youtu.be/kD2U6lghyUE.

All training data for the below experiments was generated via a custom *interactive particle-based environment* based on the FleX physics engine [31] in Unity3D. This environment provides (1) an automated way to extract a particle representation given a 3D object mesh, (2) a convenient way to generate randomized physics scenes for generating static training data, and (3) a standardized way to interact with objects in the environment through forces.[†]. Further details about the experimental setups and training procedure can be found in the supplement.

### 5.1   Qualitative evaluation of physical phenomena

*Rigid body kinematic motion and external forces.*   In a first experiment, rigid objects are pushed up, via an externally applied force, from a ground plane then fall back down and collide with the plane. The model is trained on 10 different simple shapes (cube, sphere, pyramid, cylinder, cuboid, torus, prism, octahedron, ellipsoid, flat pyramid) with 50-300 particles each. The static plane is represented using 5,000 particles with a practically infinite mass. External forces spatially dispersed with a Gaussian kernel are applied at randomly chosen points on the object. Testing is performed on

---

[†]HRN code and Unity FleX environment can be found at https://neuroailab.github.io/physics/

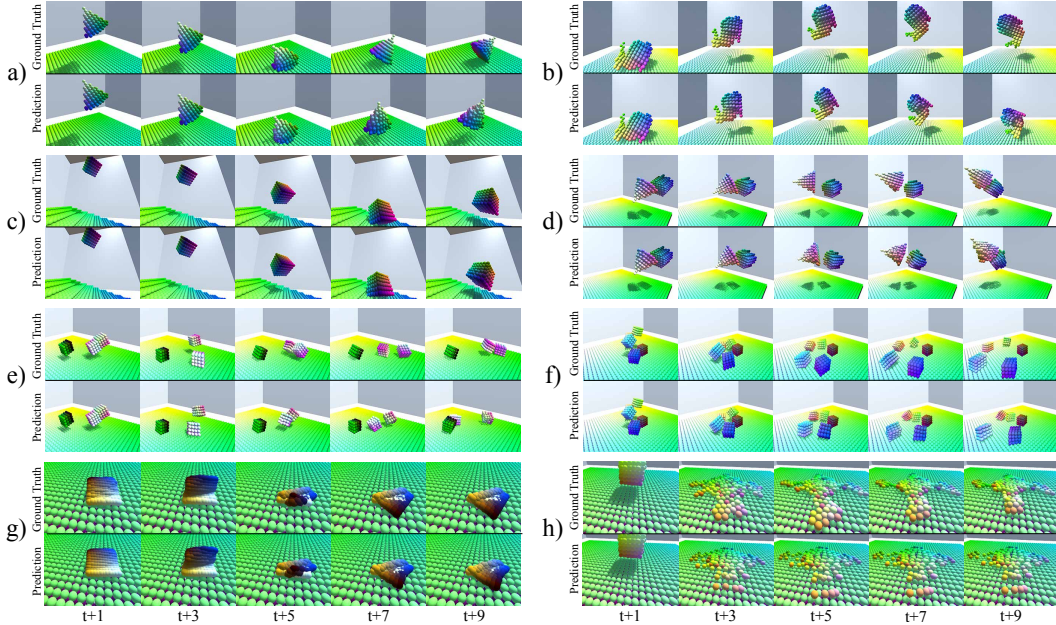

Figure 5: **Prediction examples and ground truth. a)** A cone bouncing off a plane. **b)** Parabolic motion of a bunny. A force is applied at the first frame. **c)** A cube falling on a slope. **d)** A cone colliding with a pentagonal prism. Both shapes were held-out. **e)** Three objects colliding on a plane. **f)** Falling block tower not trained on. **g)** A cloth drops and folds after hitting the floor. **h)** A fluid drop bursts on the ground. We strongly recommend watching the videos in the supplement.

instances of the same rigid shapes, but with new force vectors and application points, resulting in new trajectories. Results can be seen in supplementary Figure F.9c-d, illustrating that the HRN correctly predicts the parabolic kinematic trajectories of tangentially accelerated objects, rotation due to torque, responses to initial external impulses, and the eventual elastic collisions of the object with the floor.

***Complex shapes and surfaces.*** In more complex scenarios, we train on the simple shapes colliding with a plane then generalize to complex non-convex shapes (e.g. bunny, duck, teddy). Figure 5b shows an example prediction for the bunny; more examples are shown in supplementary Figure F.9g-h.

We also examine spheres and cubes falling on 5 complex surfaces: slope, stairs, half-pipe, bowl, and a "random" bumpy surface. See Figure 5c and supplementary Figure F.10c-e for results. We train on spheres and cubes falling on the 5 surfaces, and test on new trajectories.

***Dynamic collisions.*** Collisions between two moving objects are more complicated to predict than static collisions (e.g. between an object and the ground). We first evaluate this setup in a zero-gravity environment to obtain purely dynamic collisions. Training was performed on collisions between 9 pairs of shapes sampled from the 10 shapes in the first experiment. Figure 5d shows predictions for collisions involving shapes not seen during training, the cone and pentagonal prism, demonstrating HRN's ability to generalize across shapes. Additional examples can be found in supplementary Figure F.9e-f, showing results on trained shapes.

***Many-object interactions.*** Complex scenarios include simultaneous interactions between multiple moving objects supported by static surfaces. For example, when three objects collide on a planar surface, the model has to resolve direct object collisions, indirect collisions through intermediate objects, and forces exerted by the surface to support the objects. To illustrate the HRN's ability to handle such scenarios, we train on combinations of two and three objects (cube, stick, sphere, ellipsoid, triangular prism, cuboid, torus, pyramid) colliding simultaneously on a plane. See Figure 5e and supplementary Figure F.10f for results.

We also show that HRN trained on the two and three object collision data generalizes to complex new scenarios. Generalization tests were performed on a falling block tower, a falling domino chain, and a bowl containing multiple spheres. All setups consist of 5 objects. See Figure 5f and supplementary Figures F.9b and F.10b,g for results. Although predictions sometimes differ from ground truth in their details, results still appear plausible to human observers.

***Soft bodies.*** We repeat the same experiments but with soft bodies of varying stiffness, showing that HRN properly handles kinematics, external forces, and collisions with complex shapes and surfaces involving soft bodies. One illustrative result is depicted in Figure 1, showing a non-rigid cube as it deformably bounces off the floor. Additional examples are shown in supplementary Figure F.9g-h.

***Cloth.*** We also experiment with various cloth setups. In the first experiment, a cloth drops on the floor from a certain height and folds or deforms. In another experiment a cloth is fixated at two points and swings back and forth. Cloth predictions are very challenging as cloths do not spring back to their original shape and self-collisions have to be resolved in addition to collisions with the ground. To address this challenge, we add self-collisions, collision relationships between particles within the same object, in the collision module. Results can be seen in Figure 5g and supplementary Figure F.11 and show that the cloth motion and deformations are accurately predicted.

***Fluids.*** In order to test our models ability to predict fluids, we perform a simple experiment in which a fluid drop drops on the floor from a certain height. As effects within a fluid are mostly local, flat hierarchies with small groupings are better on fluid prediction. Results can be seen in Figure 5h and show that the fall of a liquid drop is successfully predicted when trained in this scenario.

***Response to parameter variation.*** To evaluate how the HRN responds to changes in mass, gravity and stiffness, we train on datasets in which these properties vary. During testing time we vary those parameters for the same initial starting state and evaluate how trajectories change. In supplementary Figures F.14, F.13 and F.12 we show results for each variation, illustrating e.g. how objects accelerate more rapidly in a stronger gravitational field.

***Heterogeneous materials.*** We leverage the hierarchical particle graph representation to construct objects that contain both rigid and soft parts. After training a model with objects of varying shapes and stiffnesses falling on a plane, we manually adjust individual stiffness relations to create a half-rigid half-soft object and generate HRN predictions. Supplementary Figure F.10h shows a half-rigid half-soft pyramid. Note that there is no ground truth for this example as we surpass the capabilities of the used physics simulator which is incapable of simulating objects with heterogeneous materials.

## 5.2 Quantitative evaluation and ablation

We compare HRN to several baselines and model ablations. The first baseline is a simple Multi-Layer-Perceptron (MLP) which takes the full particle representation and directly outputs the next particle states. The second baseline is the Interaction Network as defined by Battaglia et al. [4] denoted as *fully connected graph* as it corresponds to removing our hierarchy and computing on a fully connected graph. In addition, to show the importance of the $\phi_C$, $\phi_F$, and $\phi_H$ modules, we remove and replace them with simple alternatives. *No $\phi_F$* replaces the force module by concatenating the forces to the particle states and directly feeding them into $\eta$. Similarly for *no $\phi_C$*, $\phi_C$ is removed by adding the collision relations to the object relations and feeding them directly through $\eta$. In case of *no $\phi_H$*, $\phi_H$ is simply removed and not replaced with anything. Next, we show that two input time steps $(t, t-1)$ improve results by comparing it with a *1 time step* model. Lastly, we evaluate the importance of the *preservation loss* and the *global loss* component added to the *local loss*. All models are trained on scenarios where two cubes collide fall on a plane and repeatedly collide after being pushed towards each other. The models are tested on held-out trajectories of the same scenario. An additional evaluation of different grouping methods can be found in Section B of the supplement.

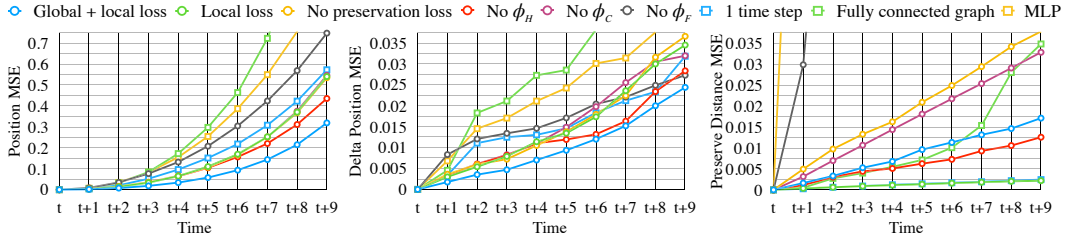

Figure 6: **Quantitative evaluation.** We compare the full *HRN (global + local loss)* to several baselines, namely *local loss only*, *no preservation loss*, *no $\phi_H$*, *no $\phi_C$*, *no $\phi_F$*, *1 time step*, *fully connected graph* and a *MLP* baseline. The line graphs from left to right show the mean squared error (MSE) between positions, delta positions and distance preservation accumulated over time. Our model has the lowest position and delta position error and a only slightly higher preservation error.

Comparison metrics are the cumulative mean squared error of the absolute global position, local position delta, and preserve distance error up to time step $t + 9$. Results are reported in Figure 6. The HRN outperforms all controls most of the time. The hierarchy is especially important, with the *fully connected graph* and *MLP* baselines performing substantially worse. Besides, the HRN without the hierarchical graph convolution mechanism performed significantly worse as seen in supplementary Figure C.4, which shows the necessity of the three consecutive graph convolution stages. In qualitative evaluations, we found that using more than one input time step improves results especially during collisions as the acceleration is better estimated which the metrics in Figure 6 confirm. We also found that splitting collisions, forces, history and effect propagation into separate modules with separate weights allows each module to specialize, improving predictions. Lastly, the proposed loss structure is crucial to model training. Without distance preservation or the global delta position prediction our model performs much worse. See supplementary Section C for further discussion on the losses and graph structures.

### 5.3 Discussion

Our results show that the vast majority of complex multi-object interactions are predicted well, including multi-point collisions between non-convex geometries and complex scenarios like the bowl containing multiple rolling balls. Although not shown, in theory, one could also simulate shattering objects by removing enough relations between particles within an object. These manipulations are of substantial interest because they go beyond what is possible to generate in our simulation environment. Additionally, predictions of especially challenging situations such as multi-block towers were also mostly effective, with objects (mostly) retaining their shapes and rolling over each other convincingly as towers collapsed (see the supplement and the video). The loss of shape preservation over time can be partially attributed to the compounding errors generated by the recursive roll-outs. Nevertheless, our model predicts the tower to collapse faster than ground truth. Predictions also jitter when objects should stand absolutely still. These failures are mainly due to the fact that the training set contained only interactions between fast-moving pairs or triplets of objects, with no scenarios with objects at rest. That it generalized to towers as well as it did is a powerful illustration of our approach. Adding a fraction of training observations with objects at rest causes towers to behave more realistically and removes the jitter overall. The training data plays a crucial role in reaching the final model performance and its generalization ability. Ideally, the training set would cover the entirety of physical phenomena in the world. However, designing such a dataset by hand is intractable and almost impossible. Thus, methods in which a self-driven agent sets up its own physical experiments will be crucial to maximize learning and understanding[19].

## 6 Conclusion

We have described a hierarchical graph-based scene representation that allows the scalable specification of arbitrary geometrical shapes and a wide variety of material properties. Using this representation, we introduced a learnable neural network based on hierarchical graph convolution that generates plausible trajectories for complex physical interactions over extended time horizons, generalizing well across shapes, masses, external and internal forces as well as material properties. Because of the particle-based nature of our representation, it naturally captures object permanence identified in cognitive science as a key feature of human object perception [43].

A wide variety of applications of this work are possible. Several of interest include developing predictive models for grasping of rigid and soft objects in robotics, and modeling the physics of 3D point cloud scans for video games or other simulations. To enable a pixel-based end-to-end trainable version of the HRN for use in key computer vision applications, it will be critical to combine our work with adaptations of existing methods (e.g. [54, 23, 15]) for inferring initial (non-hierarchical) scene graphs from LIDAR/RGBD/RGB image or video data. In the future, we also plan to remedy some of HRN's limitations, expanding the classes of materials it can handle to including inflatables or gases, and to dynamic scenarios in which objects can shatter or merge. This should involve a more sophisticated representation of material properties as well as a more nuanced hierarchical construction. Finally, it will be of great interest to evaluate to what extent HRN-type models describe patterns of human intuitive physical knowledge observed by cognitive scientists [32, 35, 38].

**Acknowledgments**

We thank Viktor Reutskyy, Miles Macklin, Mike Skolones and Rev Lebaredian for helpful discussions and their support with integrating NVIDIA FleX into our simulation environment. This work was supported by grants from the James S. McDonnell Foundation, Simons Foundation, and Sloan Foundation (DLKY), a Berry Foundation postdoctoral fellowship (NH), the NVIDIA Corporation, ONR - MURI (Stanford Lead) N00014-16-1-2127 and ONR - MURI (UCLA Lead) 1015 G TA275.

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
