[Supplementary Material]

# Supplementary Material

## A   Iterative hierarchical grouping algorithm

We describe the iterative grouping algorithm used to generate our hierarchical particle-based object representation in Algorithm 1:

---

**Algorithm 1:** Iterative hierarchical grouping algorithm.

---

**input** : Scene graph $G = <P, R>$ with particles $P$ and relations $R$ and target cluster size $N_C$
**output** : Hierarchical scene graph $G_H = <P_H, R_H>$
**begin**

    Initialize $R_H = \{\}$ and $P_H = \{\}$;

    **for** *connected component (object)* $o \in G$ **do**

        Initialize $R_o = \{\}$ and $P_o = \{p_i | i \in o\}$;

        Create root particle $p_{\text{root}} = < \frac{1}{|P_o|}\Sigma_{i \in o}x_i, \frac{1}{|P_o|}\Sigma_{i \in o}\delta_i, \Sigma_{i \in o}m_i >$ ;

        Connect $p_{\text{root}}$ to leaves($p_{\text{root}}$) with relations
        $R_{\text{A2D}} \equiv \{r_{ij} | i = \text{root}; p_j \in \text{leaves}(p_{\text{root}})\}$;

        Connect leaves($p_{\text{root}}$) to $p_{\text{root}}$ with relations
        $R_{\text{L2A}} \equiv \{r_{ij} | p_i \in \text{leaves}(p_{\text{root}}); j = \text{root}\}$;

        Add relations to $R_o \equiv R_o \cup R_{A2D} \cup R_{L2A}$;

        Initialize the particle processing queue $q = \{p_{\text{root}}\}$;

        **while** *q not empty* **do**

            Get current particle $p_{\text{curr}} = pop(q)$;

            Initialize processed subcomponent indexes $I_s = \{\}$;

            **if** $|leaves(p_{curr})| \geq N_C$ **then**

                Use k-means to group leaves($p_{\text{curr}}$) into $N_C$ subcomponents $\{S_1, S_2, ..., S_{N_C}\}$;

                **for** *subcomponent* $S \in \{S_1, S_2, ..., S_{N_c}\}$ **do**

                    **if** $|S| > 1$ **then**

                        Create new root particle for subcomponent
                        $p_s = < \frac{1}{|S|}\Sigma_{i \in S}x_i, \frac{1}{|S|}\Sigma_{i \in S}\delta_i, \Sigma_{i \in S}m_i >$ ;

                        Connect all anc($p_s$) to $p_s$ with relations $R^1_{\text{A2D}} \equiv \{r_{is} | p_i \in \text{anc}(p_s)\}$ ;

                        Connect $p_s$ to all leaves($p_s$) with relations $R^2_{\text{A2D}} \equiv \{r_{sj} | p_j \in \text{leaves}(p_s)\}$;

                        Connect all leaves($p_s$) to $p_s$ with relations $R_{\text{L2A}} \equiv \{r_{is} | p_i \in \text{leaves}(p_s)\}$;

                        Add relations to $R_o \equiv R_o \cup R^1_{\text{A2D}} \cup R^2_{\text{A2D}} \cup R_{L2A}$;

                        Add $p_s$ to $P_o \equiv P_o \cup \{p_s\}$;

                        Add $s$ to $I_s \equiv I_s \cup \{s\}$;

                        Append $p_s$ to processing queue $q = push(p_s, q)$;

                  **end**

                  **else**

                    Add $S$ to $I_s \equiv I_s \cup S$;

                  **end**

                **end**

            **end**

            **else**

                Set $I_s \equiv \{i | p_i \in \text{leaves}(p_{\text{curr}})\}$;

            **end**

            Connect all particle pairs $p_i$ and $p_j$ in $I_s$ with $R_{\text{WS}} \equiv \{r_{ij} | i, j \in I_s\}$ ;

            Add $R_{\text{WS}}$ to $R_o \equiv R_o \cup R_{\text{WS}}$ ;

        **end**

        Add relations $R_o$ to $R_H \equiv R_H \cup R_o$ ;

        Add particles $P_o$ to $P_H \equiv P_H \cup P_o$;

    **end**

    Return $G_H = <P_H, R_H>$;

**end**

---

# B    Comparison of different grouping methods

While performing a hyperparamter search we also tried several different grouping methods. Here, we compare agglomerative clustering against different versions of k-means. Specifically, we tried to generate hierarchies with up to 8 particles and 10 particles per group grouped by k-means. As seen in Figure B.1 and Figure B.2 we found that k-means with 8 particle groups works best resulting in a reasonable trade-off between number of particles per group and number of hierarchical layers for the tested objects. However, the improvement over the other clustering algorithms is minor, indicating that HRN is robust to the grouping method.

Figure B.1: **Qualitative comparison of different grouping methods.** Agglomerative grouping (top) is compared against k-means with up to 10 particles per group (middle), and k-means with up to 8 particles per group (bottom) which is used in HRN.

Figure B.2: **Quantitative comparison of different grouping methods.** Agglomerative grouping (yellow) is compared against k-means with up to 10 particles per group (green), and k-means with up to 8 particles per group (blue) which is used in HRN.

# C    Comparison of different losses and graph structures

This section complements the quantitative evaluation and ablation studies. Figure C.3 compares the predictions of models trained with the different loss terms. Results of a model trained with a combination of global and local losses are visually closest to ground truth. These qualitative results align well with the quantitative results in Figure C.4 and Figure 6.

Figure C.4 also illustrates the importance of a hierarchical graph (global + local loss) compared to a sparse flat graph or a fully connected graph. While the fully connected graph performs worse than the sparse flat graph and the hierarchical graph on all metrics, the sparse flat graph is comparable to the hierarchical graph on the position and delta position MSE. However, the sparse flat graph does much worse on the preserve distance MSE, indicating that the original object shape is hardly preserved. Presumably, the effect propagation in the sparse flat graph is less effective than in the hierarchical graph leading to acceptable particle positions but deformed objects.

Summarizing, a better performance on the quantitative metrics (position MSE, delta position MSE and preserve distance MSE) indeed results in qualitatively better examples. Our final combination of global and local loss terms outperforms each individual loss on its own. Similarly, our hierarchical graph significantly improves predictions compared to a sparse flat graph or a fully connected graph.

Figure C.3: **Qualitative comparison of different loss terms.** Combining global and local loss terms (top) results in predictions closest to the ground truth (bottom) compared with using no preservation loss, a local loss or global loss by itself.

Figure C.4: **Quantitative comparison of different losses and graph structures.** Losses and graph structure are ablated from left to right. In terms of losses, the full HRN with global + local loss (blue) is compared against local loss only (green), global loss only (red) and a loss without a preserve distance term (yellow). Regarding graph structure, the full HRN (blue) is compared against a sparse flat graph in which the hierarchy was removed (purple) and a fully connected graph structure (black) as presented in Battaglia et al. [4].

# D  Implementation details

## D.1  Detailed model structure

The HRN is given the states $P_o^{(t-T,t]}$, the gravity $g$ and any external forces $F$. It is trained to predict the future particle states $P_o^{t+1}$ for each object $o$. In our implementation, the model actually predicts the change in local position $\Delta^{t+1} \equiv X^{t+1} - X^t$, and use $\Delta^{t+1}$ to advance the particle states. Note that $X_o^t \equiv \{x_j^t | p_j \in P_o\}$ is the set of all particle positions in $o$.

Figure D.5 shows a detailed overview of HRN model architecture. In total, there are five modules, each with their own MLP. The dotted box denotes shared weights between the three hierarchical graph convolution stages, $\eta_{L2A}$, $\eta_{WS}$, and $\eta_{A2D}$. All MLPs use a ReLU nonlinearity. The number of units, layers, and output dimension of each MLP were chosen through a hyperparameter search. The gravity input $g$ to $\Psi$ is only added for the global super-particles of each object.

## D.2  Training procedure

We train the network using the Adam optimizer with a batch size of 256 across multiple Nvidia Titan Xp GPUs. The initial learning rate was set at 0.001 and decayed stepwise a total of 3 times, alternating between a factor of 2 and 5 each step. We used TensorFlow for the implementation. For the generalization experiments we include data augmentation in the form of random grouping, mass, and translation.

Figure D.5: **Detailed description of the HRN model architecture.**

# E   Detailed experimental setups

## E.1   Particle-based physics simulation environment

Based on the FleX physics engine [31] we built a custom *interactive particle-based environment* in Unity3D. This environment automatically decomposes any given 3D object mesh into a particle representation using the FleX API. On top of this representation it provides a convenient way to generate randomized physics scenes for generating static training data. The user is able to construct random scenes through a python interface that communicates with Unity3D. This interfaces also

allows for physical interactions with objects within a defined scene. For instance, one can apply forces to a whole object or individual particles to generate translational and rotational position variations. It is both possible to generate static datasets from the environment and to train offline as well as to train and interact with the environment online. Therefore the environment sends the python script client the particle state at every frame as well as images captured by a camera in the scene. Scenes can be rendered with around 30 frames per second. The simulation time increases with the number of particles. Figure E.6 shows a screenshot of the environment embedded in the Unity3D editor. Mesh skins are used to mask the particles in the main scene to give the impression of a continuous object. In the lower right of this screenshot we can see the particle representation of the cube in the scene after FleX has converted the 3D mesh into a particle representation. Code for this environment, along with the entire HRN code base, can be found at `https://neuroailab.github.io/physics/`.

Figure E.6: **Particle-based Interaction Environment in Unity3D.** Screenshot of the Unity Editor with FleX Plugin. In the main scene a cube is colliding with a planar surface. The lower right shows the particle representation of the cube. This environment is used to generate training and validation data through interactions with objects in the scene. Interactions with the environment are possible through a python interface.

## E.2 Shapes and surfaces used during experiments

Figure E.7 and Figure E.8 show the 3D mesh and the leaf particle representation of all shapes and surfaces used during training or testing. Moving objects consist of 50-300 particles, surfaces of more than 5000 particles. Only one particle resolution is shown although multiple levels of detail in the leaf node representation are possible by changing the particle spacing within an object.

## E.3 Throwing one object in the air

In this experiment any one of the small shapes depicted in Figure E.7 is first chosen to collide with one of the surfaces in Figure E.8. The small shape is teleported to a random location around the center of the surface. The stiffness is randomly chosen per object after a teleport. As the simulation starts the shape falls on the surface and collides with it. Every random number of frames we apply a randomly upward and perpendicular to the surface pointing force to lift the object up and watch it fall again as it describes a parabola. If the object leaves the surface boundaries we randomly teleport it back to the center. After a fixed number of steps we reinitialize the scene and the whole simulation procedure starts again.

Figure E.7: **Dynamic shapes and particle representations.** All shapes used during testing and training are shown. Shapes consist of 50 - 300 particles. Only one particle resolutions is shown.

Figure E.8: **Surfaces and particle representations.** All surfaces used during testing and training are shown. Surfaces consist of 5000 - 7000 particles.

## E.4 Cloths

Two different experiments are performed to test our model on predicting the motion of a cloth. The first experiment is similar to *throwing an object in the air*. A loose cloth is teleported to a random location above the ground. On simulation start the cloth drops on the ground. Then, every fixed number of frames we apply a random force dispersed by a Gaussian kernel to the cloth and watch it deform. After a fixed number of steps we reinitialize the scene and the whole simulation procedure starts again. In the second experiments, we attach two corners of the cloth to a random location in the

air. Every fixed number of steps a random force is applied to the cloth which deforms the cloth and makes it swing back and forth. The scene is reset after a fixed number of frames and the two cloth corners are attached at a new random location.

## E.5 Fluids

In the fluid experiment a cube shaped fluid is teleported to a random location around the center of the ground. As the simulation starts the fluid drops on the ground and disperses on contact with the ground. The fluid's surface tension holds it together such that fluid particles cluster in one or few water puddles. After a set number of frames the fluid is reset to its original cube-like shape and teleported to the next random location.

## E.6 Collisions between objects without gravity

This experimental setup is very similar to *throwing an object in the air* with the difference that gravity is disabled, and we choose two small dynamic shapes that collide with each other in the air. The stiffness is randomly chosen per object after a teleport. Forces are applied such that they either point directly from one object to the other or away from each other. The force magnitude and perturbations to the force direction are randomly chosen every time an action is applied. Forces are applied randomly either to one or both objects at the same time. The simulation is reinitialized if any of the two objects leaves the room boundaries.

## E.7 Collisions between objects on a planar surface

This experiment is a combination of the previous two experiments. Just as in *throwing one object in the air* the two or three chosen small objects are spawned randomly around the center of the planar surface. The stiffness is randomly chosen per object after a teleport. They fall and collide with the plane. Similar to *collisions between objects without gravity* the force is applied such that the two objects collide with each other or are torn apart. The force magnitude and perturbations to the force direction are chosen randomly. Forces are applied randomly either to one or two objects at the same time. The scene is reinitialized if any of the two objects leaves the surface boundaries.

## E.8 Stacked tower

In this experiment we manually construct a tower consisting of 5 stacked rigid cubes on a planar surface. The positions of the cubes are slightly randomly perturbed to create towers of variable stability. After a random number of frames a force is applied to a randomly chosen cube which is usually big enough to make the tower fall. Once the tower falls and the cubes do not move anymore or after a maximum number of time steps the setup is reset and repeated.

## E.9 Dominoes

Similar to the stacked tower, we manually setup a scene in which a rigid dominoes chain is placed on top of a planar surface. Small random perturbations are applied to the initial position of each domino. After a random number of frames a force is applied to one or both sides of the chain to make it fall. Once dominoes do not move anymore or after a fixed maximum number of time steps the setup is reset and repeated.

## E.10 Balls in bowl

The last manually constructed control example are 5 balls dropping into a big bowl. The spheres are teleported to a randomly chosen position above the bowl. The balls then drop into the ball and interact with each other. A random force is applied every random number of frames. Once the spheres have settled or after a maximum number of time steps we reinitialize the scene.

# F   Qualitative prediction examples

This section showcases additional qualitative prediction examples. Figure F.9 and Figure F.10 show additional examples with different objects and physical setups and failure cases. Figure F.11 visualizes additional cloth predictions.

In Figure F.12 we demonstrate the model's ability to handle varying stiffness inputs. The network is trained on multiple soft bodies of varying stiffness. The stiffness values are obtained from FleX during dataset generation and vary between 0.1 and 0.9 for soft bodies. By manually changing the input stiffness during testing, we can produce predictions of objects with varying levels of rigidity. The decreasing level of deformation in frame $t + 5$, from top to bottom, is consistent with the increasing stiffness.

We also test whether the model can capture physical relationships in varying gravitational fields. Since the value of gravity is also an input to our model, we can train on data with a changing gravitational constant. Figure F.13 shows an example with four different gravitational constants, ranging from 1 to $20 \ m/s^2$. As expected, the object falls faster with more gravity.

As part of the particle state, we include the mass of each particle. While the total object mass is usually kept constant for most of the experiments, we test the case of varying mass by training on a dataset where the each object's mass will vary by a factor of up to three times. In Figure F.14 we manually increase the mass of one of the two objects in the collision and show that the heavier object is displaced less after the collision.

Figure F.9: **Qualitative comparison of HRN predictions vs ground truth. a)** A sphere falling out of a bowl. Objects containing other objects can be easily modeled. **b)** Five spheres fall into a ball and collide with each other. Complex indirect collisions occur. **c)** A rigid pyramid colliding with the floor. **d)** A rigid sphere colliding with the floor. **e)** A cylinder colliding with a pyramid. **f)** Ellipsoid and octahedron colliding with each other. **g)** A soft teddy colliding with the floor. **h)** A soft duck colliding with the floor.

Figure F.10: **Qualitative comparison of HRN predictions vs ground truth. a)** A very deformable stick. The ground truth shape had to be fed into the model for this prediction to work. **b)** Falling dominoes. HRN wrongly predicts one domino moving off to the side in this complex multi-object interaction scenario. **c)** A rigid cube colliding with stairs. **d)** A cube colliding with a random surface. **e)** A ball on a slope. **f)** Three objects colliding with each other. **g)** A slowly falling tower. The tower in the HRN prediction collapses much faster compared to ground truth. **h)** A half-rigid (right object side) half-soft (left object side) body colliding with a planar surface. The soft part deforms. The rigid part does not deform.

Figure F.11: **Qualitative comparison of HRN predictions vs ground truth. a)** Dropping cloth. Cloth drops from a certain height onto the ground. **b)** Hanging cloth. Cloth is fixated at two points and swings back and forth.

Figure F.12: **Responsiveness to stiffness variations.** We vary the stiffness of a cube colliding with a planar surface. The top row shows a soft cube with stiffness value 0.1, the middle row a stiffness value of 0.5, and the bottom row a almost rigid cube with a stiffness value of 0.9. Our network responds as expected to the changing stiffness value deforming the soft cube stronger than the rigid cube.

Figure F.13: **Responsiveness to gravity variations.** We vary the gravity while a soft cube is falling on a planar surface. From top to bottom gravity values of 1.0, 5.0, 10.0 and 20.0 $m/s^2$ are depicted. We can see that the cube falls faster as gravity increases and even deforms the object when colliding with the floor under strong gravity. Our model behaves as expected when gravity changes.

Figure F.14: **Responsiveness to mass variations.** We vary the mass while two cubes collide. The top shows a scenario where the purple cube is heavy. Here the green cube bounces off stronger than the purple one. The bottom shows the same scenario but with green cube being heavy. Here the green cube doesn't move as strongly.