[Reviews · NeurIPS 2018]

Reviewer 1



This work extends the work in Battaglia et al., of learning the dynamics of physical systems using leanred message passing, to large systems. It is an important and interesting work and mostly well written, although some key points need clarification. Detailed remarks: - In line 52 you mention how a full graph scales badly (and in other locations in the paper), but there is the option of a sparse graph, e.g. only KNN edges. It would be important to compare and contrast this alternative. - Your clustering can introduce boundary effects (which might be the reason behind your sub-par perserve distance MSE), it should be discussed and an empirical evaluation (e.g. how much of the per-dist MSE is between clusters vs in clusters) is in place. - The high level part of the hierarchical effect propagation isn’t that clear. The way I understand it is that you compute e^0 on leaves, then send e^L2A up the hierarchy till the root, then compute e^WS at each level and finally compute w^A2D down the hierarchy to the leaves. If this is true it would be helpful to say that in the paper, if not then it is not clear enough and needs clarification. - If it wasn’t for the consecutive message passing, it would be equivalent to a graph neural network with 3 edge types (L2A,A2S and WS), would be useful to compare. Minor remark: - Line 94 “,excluding e.g. …” sounds bad. “Excluding, for example, …” is a better option. Update: Thank you for your response. Good point about kNN needing multiple rounds (at least the naive way).

Reviewer 2



The authors propose a novel hierarchical object representation based on particles to cover both rigid geometrical shapes and deformable materials. Each scene is represented as a graph, with disconnected components corresponding to the objects and the support of the scene. Each graph has a tree-like structure, where higher levels correspond to coarser scales, and the leaves correspond to the original particles placed in the object. They also propose an adapted neural network architecture, called Hierarchical Relation Network, that learns to predict physical dynamics for this representation. This architecture propagates exterior effects (forces, collisions) as well as the effect of past states through the graph, first from the leaves to their ancestors, then among sibling nodes, and finally back to the leaves. This multiscale approach is end to end differentiable, allowing this propagation mechanism to be learned. The approach is very nicely motivated, and addresses an important problem: learning to model and predict physical states and dynamics of complex objects, and their interactions, in an end to end learnable fashion, so that it could be integrated in larger learning systems. This could be very useful in scenarios of computer vision, (eg. for action recognition tracking and future prediction), robotics (action conditioned future prediction is crucial for navigation) and reinforcement learning. Overall the paper is well written. The authors cite and compare to the most relevant prior work in a convincing manner - although it would be worthwhile citing "Learning Physical Intuition of Block Towers by Example" by Lerer et al. as well as "IntPhys: A Benchmark for Visual Intuitive Physics Reasoning" by Riochet et al. The approach is novel, and the experimental set up they propose is challenging, and would certainly serve future research purposes. In terms of weaknesses of the approach, the proposed message passing scheme, relying on the hierarchical representation, does not seem very principled. It would be nice to know more precisely what has inspired the authors to make these design choices. Perhaps as a consequence, in qualitative results show sometimes objects seem to rip appart or smoothly deform into independent pieces. Qualitatively, some results are impressive, while some do not conform to our expectations of what would happen; cf my previous comment on unexpected deformations of objects, but also some motions: cubes moving slightly on the ground although they should be still, etc. More importantly, the applicability of the method is questionable, as it requires ground truth for the local and global deltas of each particle for training. Furthermore, even for inference, it would require the initial states of the particles. This would be a huge challenge in computer vision applications, and in any other applications relying on real videos, rather than synthetic videos. This greatly limits the applicability of the proposed representation, and as a consequence of the proposed architecture. Therefore I have doubts at this point that the proposed architecture has "the potential to form the basis of next generation physics predictors for use in computer vision, robotics, and quantitative cognitive science". To claim this, the authors should be able to show preliminary results of how their method could be used, even in very simple, but realistic settings. Nevertheless I believe that it is worthwhile sharing these results with the community, as well as the code for the method and the environment, as this is an important problem, and the authors attack it with an ambitious and novel (although slightly ad hoc) approach. To be fully confident that this paper should be accepted, I would like the authors to clarify the following points: - it is not clear how the quantitative results are obtained: what data exactly is used for training, validating and testing ? - ternary input to the phi network, encoding the type of pairwise relationship -> is this effective ? What happens otherwise, if this is not given, or if weights are not shared ? Also wouldn't it make sense to have categorical input in the form of a one hot vector instead ? - the algorithm described for the creation of the graph G_H is inconsistent with the description of the remaining edges (l.122 to 1.131). According to the algorithm, there should only be edges between siblings, from the root to the leaves and back, and from the root to all intermediate nodes. Maybe this is due to l.127: "each new node taking the place for its child leaves" should be "each new node taking the place of the root" ? This would then also be more consistent with Algorithm 1. Please clarify. In Figure 3b) it is not clear what the red links represent, if the edges are supposed to be directed. - I don't understand the process for learning the r_ij. If it can be different for each pair of particle (or even for each source particle i), how can it be learned using the losses described in 4.3 ? We can obtain their values on the objects of the training set, but then for each new object, we need to gather synthetic videos of the object moving, in order to learn its properties? Wouldn't it be simpler to have fixed parameters ? Furthermore, if the stiffness of the material is to be learned, why not learn the mass as well ? - in the algorithm described for the creation of the graph G_H, it is said that one should "add edges between cluster nodes if the clusters are connected by edges between leaves" - how is there a presence or an absence of an edge in the first place ? Nitpicking: - are humans really able to predict subtle deformations of masses at different scales ? Or can they simply perceive them and say whether they are plausible or not ?(l.39-44) Typos: - par(i) should be par(p) l.128 - Figure 3: "is constraint" should be "is constrained" - l.152; l should be p_l ? - l 155 repetition with the following: "Effects are summed together" -------------------------------- Final decision after rebuttal: The authors have addressed all of my concerns, except for one which I share with R3 that the experimental protocol for the quantitative results is not described. Overall however I think that the paper presents an ambitious and novel method to a very important problem, which they convincingly test in a challenging experimental set up. I agree with R3 that the qualitative evaluation should include the weaknesses in the main paper and I think it is beneficial that the authors have agreed to do so. I remain convinced that it would be worth sharing these results with the community. I can only urge the authors to respect their commitment to share the code and environment, and to add the missing details about the experimental protocol, to ease further research in this area and comparison to their work. I stand by my initial decision that the paper should be accepted as the approach is a very well executed stepping stone for others to improve on in the extremely challenging setting proposed by the authors.

Reviewer 3



The submission presents a hierarchical particle-based object representation that acts as a physics prediction engine for 3-dimensional deformable bodies. The paper is strongly based off Battaglia et al. work on 2D physics prediction. To handle the 3D physical structure of the observed objects, the objects are approximated by a set of particles. The main contribution of this submission is to propose a hierarchical particle structure that can propagate external forces on throughout the entire body. The neural network architecture is described, which consists of separate hierarchical graph convolution modules (one each to capture the effect of external forces, collisions and past states) that are summed up and fed into another graph neural network and a fully connected layer. A custom loss function is presented that consists of local positional difference, global positional difference and difference in intra-object particle structure. Strengths of this submission: The author tries to find a solution to an interesting problem: Can a machine learn to predict physical behavior of objects such as collisions or material behavior, without knowledge of any physics laws or formula? The submission is well structured and explains a complex model well and in detail. Some of the results show a realistic prediction of the physical behavior. The authors present three different metrics that can be used to quantitatively compare different models with each other. Weaknesses of this submission: The qualitative evaluation of physical phenomena seems too optimistic. Although some results are very impressive, the prediction lacks in more complex problems such as multi body collisions. Weaknesses are shown in additional results, but not discussed in the main paper. The quantitative evaluation is somewhat lacking, in that it is not explained which experiment the results are based on. Furthermore, Figure 6 seems to show that there is a trade off in using local loss function vs. global + local loss function, in that only local loss function seems to perform better on preserve distance error. This trade off is not discussed in the paper (“Without […] the global delta position prediction our model performs much worse”). In general, I would have preferred to see a more detailed discussion of simpler cases (single body with external forces and collision with ground), i.e. showing qualitatively that physical properties like rigidity, friction, gravitation etc. can be learned without knowledge of the governing laws of physics. I thank the reviewers for their detailed response.